# SpeechOp: Inference-Time Task Composition for Generative Speech Processing

**Justin Lovelace**[*]
Cornell University
jl3353@cornell.edu

**Rithesh Kumar  Jiaqi Su  Ke Chen**
Adobe Research
{ritheshk,jsu,kechen}@adobe.com

**Kilian Q Weinberger**
Cornell University
kqw4@cornell.edu

**Zeyu Jin**
Adobe Research
zejin@adobe.com

## Abstract

While generative Text-to-Speech (TTS) systems leverage vast "in-the-wild" data to achieve remarkable success, speech-to-speech processing tasks like enhancement face data limitations, which lead data-hungry generative approaches to distort speech content and speaker identity. To bridge this gap, we present SpeechOp, a multi-task latent diffusion model that transforms pre-trained TTS models into a universal speech processor capable of performing a wide range of speech tasks and composing them in novel ways at inference time. By adapting a pre-trained TTS model, SpeechOp inherits a rich understanding of natural speech, accelerating training and improving S2S task quality, while simultaneously enhancing core TTS performance. Finally, we introduce Implicit Task Composition (ITC), a novel pipeline where ASR-derived transcripts (e.g., from Whisper) guide SpeechOp's enhancement via our principled inference-time task composition. ITC achieves state-of-the-art content preservation by robustly combining web-scale speech understanding with SpeechOp's generative capabilities. Audio samples are available at https://justinlovelace.github.io/projects/speechop.

## 1 Introduction

Generative Text-to-Speech (TTS) systems now produce increasingly natural and expressive speech (Le et al., 2024; Ju et al., 2024), largely due to their ability to leverage vast "in-the-wild" data (e.g., from audiobooks, podcasts (Chen et al., 2021a; Pratap et al., 2020)). This scalability enables TTS models to learn robust speech representations across diverse acoustic conditions and speaker characteristics (Lee et al., 2024; Peng et al., 2024).

In contrast, speech-to-speech (S2S) processing tasks like enhancement, speaker separation, and foreground-background isolation face stricter data requirements, often needing paired degraded/clean speech, which is expensive to acquire at scale (Zen et al., 2019). Consequently, S2S models are typically trained on smaller, specialized datasets, often with simulated degradations (Su et al., 2021a). This data scarcity can cause generative S2S approaches to distort original speaker identity and content—a critical issue where faithful preservation is paramount, e.g., in speech enhancement (Yang et al., 2024; Koizumi et al., 2023b). These models often lack the rich speech understanding derived from vast, diverse datasets available to TTS.

To bridge this data gap, we present SpeechOp: a multi-task latent diffusion model that transforms pre-trained TTS models into a universal speech processor. SpeechOp performs a wide range of S2S tasks and allows their novel inference-time composition, leading to three key advancements (Figure 1):

**1. A Flexible Multi-Task Model That Enhances TTS Capabilities:** SpeechOp, adapted from a pre-trained TTS model and fine-tuned on diverse S2S tasks (including TTS, enhancement, separation),

---

[*]Work done during an internship at Adobe.

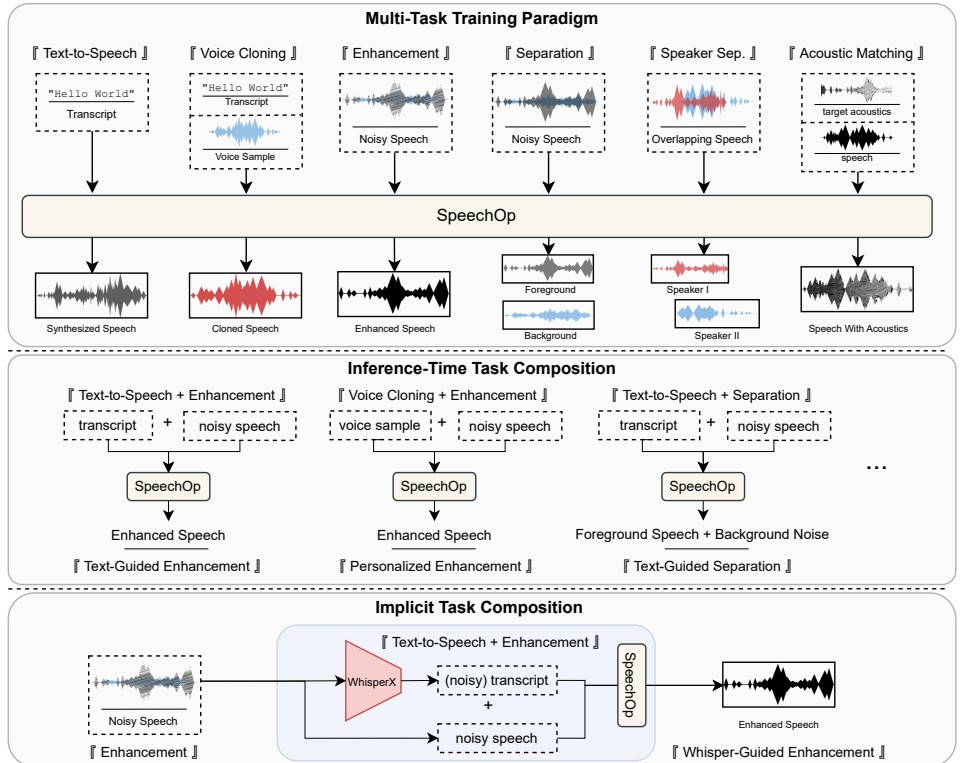

Figure 1: Overview of SpeechOp's multi-task training (top), inference-time task composition capabilities (middle), and implicit task composition pipeline (bottom). The model is trained on six core speech tasks including text-to-speech, enhancement, and separation. At inference time, novel tasks can be composed by combining learned capabilities - for example, using transcripts to guide enhancement or personalizing enhancement with voice samples. In the implicit task composition pipeline, we use a state-of-the-art discriminative model (Whisper) to automatically transcribe noisy speech, then use the resulting transcript to guide SpeechOp's enhancement process.

not only becomes a versatile speech processor but also *improves* its underlying TTS quality. By learning to handle varied acoustic manipulations, SpeechOp's TTS component generates more natural, higher-quality speech, validated by human listening studies.

**2. Inference-Time Task Composition (TC-CFG):** Our novel TC-CFG guidance strategy (Section 6) enables composing independently trained speech capabilities at inference time without requiring joint training. For instance, SpeechOp can combine enhancement with TTS content guidance to both improve acoustics and re-synthesize obscured content.

**3. State-of-the-Art Speech Processing through Implicit Task Composition (ITC):** SpeechOp achieves state-of-the-art content preservation via ITC. Traditional transcript-conditioned S2S models suffer from scarce paired noisy-clean-transcript data and the propagation of ASR errors. ITC overcomes these by robustly integrating ASR-derived transcripts (e.g., from Whisper (Radford et al., 2023; Bain et al., 2023)) using our TC-CFG inference-time composition. This principled approach, with its tunable "guidance strength," allows balancing content restoration (more like TTS) and acoustic fidelity (more like enhancement) based on the situation, achieving superior content fidelity over specialized enhancement methods.

## 2 BACKGROUND: DIFFUSION MODELS

We introduce latent diffusion models following recent formulations (Ho et al., 2020; Kingma & Gao, 2023; Rombach et al., 2021). Given data drawn from an unknown distribution $q(\mathbf{x})$, our goal is to learn a generative model $p_\theta(\mathbf{x})$ that approximates this distribution.

**Forward process.** The forward process defines a gradual transition from the latent distribution to a Gaussian distribution through a sequence of increasingly noisy latent variables $\mathbf{z}_t$ for timesteps $t \in [0, 1]$. This Gaussian diffusion process defines the conditional distribution $q(\mathbf{z}_{0,\dots,1}|\mathbf{x})$. For every $t \in [0, 1]$, the marginal $q(\mathbf{z}_t|\mathbf{x})$ is given by:

$$\mathbf{z}_t = \alpha_t \mathbf{x} + \sigma_t \boldsymbol{\epsilon}, \quad \text{where} \quad \boldsymbol{\epsilon} \sim \mathcal{N}(\mathbf{0}, \mathbf{I})$$

We use the variance-preserving formulation where $\sigma_t^2 = 1 - \alpha_t^2$. The noise schedule $\alpha_t \in [0, 1]$ is a strictly monotonically decreasing function that starts with the original latent ($\mathbf{z}_0 \approx \mathbf{x}$) and ends with approximately Gaussian noise ($q(\mathbf{z}_1) \approx \mathcal{N}(\mathbf{z}_1; \mathbf{0}, \mathbf{I})$).

**Generative model.** Given the score function $\nabla_{\mathbf{z}} \log q_t(\mathbf{z})$, or the gradient of the log probability density function, we can reverse the forward process exactly. Diffusion models utilize a neural network to learn to estimate the score function, $\mathbf{s}_\theta(\mathbf{z}; \lambda) \approx \nabla_{\mathbf{z}} \log q_t(\mathbf{z})$, and use the estimated score function to approximately reverse the forward process. If $\mathbf{s}_\theta(\mathbf{z}; \lambda) \approx \nabla_{\mathbf{z}} \log q_t(\mathbf{z})$, then our generative distribution is close to the true distribution. This enables us to draw samples from a Gaussian distribution $\mathbf{z}_1 \sim p(\mathbf{z}_1)$, and approximately solve the reverse diffusion process using the estimated score $\mathbf{s}_\theta(\mathbf{z}; \lambda)$.

**Training objective.** We train the score network using a denoising score matching (DSM) loss Song & Ermon (2019) over all data points $\mathbf{x} \sim \mathcal{D}$ and noise levels:

$$\mathcal{L}_{\text{DSM}}(\mathbf{x}) = \mathbb{E}_{t,\mathbf{x},\boldsymbol{\epsilon}}[w(\lambda_t) \cdot \|\mathbf{s}_\theta(\mathbf{z}_t; \lambda) - \nabla_{\mathbf{z}_t} \log q(\mathbf{z}_t|\mathbf{x})\|_2^2],$$

where $w(\lambda_t)$ weights different noise levels during training. Following best practices (Salimans & Ho, 2022), we adopt the velocity parameterization, $\mathbf{v} = \alpha_t \boldsymbol{\epsilon} - \sigma_t \mathbf{x}$, for our network output to ensure training stability.

## 3 RELATED WORK

Text-to-speech (TTS) systems (Le et al., 2024; Shen et al., 2024) excel due to vast "in-the-wild" data, unlike data-limited speech-to-speech (S2S) tasks like enhancement (Koizumi et al., 2023b) and separation, which often require scarce paired recordings. While multi-task autoregressive models have been developed (Wang et al., 2024), they lack inference-time compositionality. Diffusion models offer high-quality synthesis (Shen et al., 2024; Le et al., 2024) and are inherently compositional (Liu et al., 2022), which SpeechOp leverages.

SpeechOp adapts pre-trained TTS models for S2S tasks and utilizes a novel inference-time composition pipeline to significantly improve speech processing quality and flexibility. Recent work like Fugatto (Valle et al., 2025) also explores multi-task audio generation. However, SpeechOp's principled task composition (TC-CFG, Section 6) provides superior control and performance in S2S tasks like enhancement compared to Fugatto's score averaging approach (Section 8), enabling effective combination of operations like enhancement and TTS. While foundational models like UniAudio (Yang et al., 2023) pursue broad task coverage from scratch, and SpeechFlow (Liu et al., 2024) investigates new pre-training schemes, SpeechOp focuses on efficiently adapting **existing** TTS models. Crucially, we introduce Implicit Task Composition (ITC), which uniquely integrates ASR models (e.g., Whisper) via TC-CFG for robust content preservation. Our primary aim is not maximizing task variety, but demonstrating how TTS pre-training and sophisticated composition can address S2S data scarcity and improve performance on established operations.

## 4 TTS PRE-TRAINING IMPROVES SPEECH PROCESSING TASKS

To motivate our multi-task framework, we first examine the benefits of initializing single-task speech enhancement and speaker separation models from a pre-trained DiT TTS backbone (Peebles & Xie, 2023; Lee et al., 2024). Figure 2 (Left) shows that TTS initialization dramatically accelerates convergence, achieving comparable validation loss with $4\times$ fewer steps for enhancement and $8\times$ fewer for separation versus random initialization. The significant speedup for separation, a complicated multi-speaker task, demonstrates the strong positive transfer from TTS pre-training.

Beyond faster training, TTS pre-training yields downstream performance gains (Figure 2 Right). Speaker separation benefits most, with TTS initialization leading to dramatic improvements in MCD

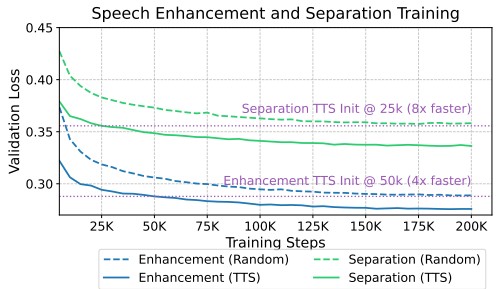

| | | Speaker Separation | | | |
|---|---|---|---|---|---|
| Init | SI-SDRi ↑ | MCD ↓ | SpBS ↑ | WER ↓ | Val MSE ↓ |
| Rand | -3.99 | 22.95 | .825 | 17.8 | 0.358 |
| TTS | -1.38 | 4.46 | .906 | 8.5 | 0.336 |

| | | Speech Enhancement | | | |
|---|---|---|---|---|---|
| Init | PESQ ↑ | MCD ↓ | SpBS ↑ | WER ↓ | Val MSE ↓ |
| Rand | 2.07 | 4.76 | .900 | 8.1 | 0.289 |
| TTS | 2.10 | 4.69 | .910 | 8.1 | 0.276 |

Figure 2: Impact of TTS initialization on speech processing tasks. (**Left**) Validation loss curves demonstrating accelerated convergence with TTS initialization. Training time is reduced by $4\times$ for enhancement and $8\times$ for separation. (**Right**) Performance metrics for speaker separation and speech enhancement, comparing random initialization (Rand) with TTS initialization (TTS).

and WER (8.5% vs 17.8%). We observe that it eliminates artifacts present in randomly initialized models that struggle to learn the content disentanglement objective. Speech enhancement also sees improvements in PESQ, MCD, and SpeechBERTScore from the TTS initialization. These results demonstrate the broad advantages of TTS pre-training—accelerated convergence and enhanced performance across diverse S2S tasks, especially those requiring deep speech understanding. This motivates SpeechOp, our multi-task framework leveraging TTS pre-training for high-quality, versatile speech processing.

## 5 SPEECHOP

We present the tasks explored in this work in Figure 1. These tasks provide complementary capabilities that are composable via our diffusion framework for applications like transcript-guided isolation. For speaker separation, which requires a speaker prompt to identify the target speaker, we provide a disjoint speech sample to disambiguate the target speaker. For foreground/background separation, we parameterize them as two separate tasks.

**SpeechOp Architecture.** SpeechOp is built on a latent diffusion framework (Rombach et al., 2021) that operates with compressed audio representations. Rather than working with raw waveforms, we first compress the audio using a DAC variational autoencoder Kumar et al. (2023) (details in Appendix), allowing our model to efficiently process and generate speech in a lower-dimensional latent space. As shown in Figure 3, SpeechOp's core architecture consists of a Diffusion Transformer (DiT) (Peebles & Xie, 2023) that is extended to handle both text-to-speech and speech-to-speech tasks. The model processes text transcripts for TTS and source audio (like noisy speech) for speech-to-speech tasks, with a learnable Task Embedding that conditions model behavior. Training proceeds in two stages: TTS pre-training followed by multi-task training to enable speech-to-speech capabilities.

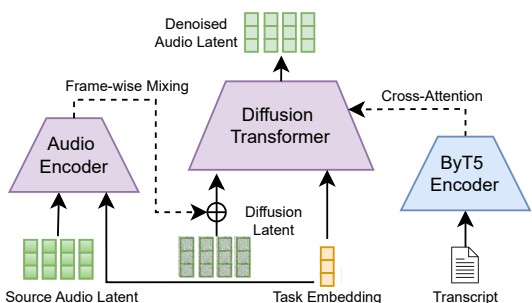

Figure 3: SpeechOp Architecture Overview. A learnable Task Embedding conditions both the Audio Encoder and Diffusion Transformer via adaptive normalization layers (Peebles & Xie, 2023) to specialize behavior for each task.

**Text-to-Speech Pathway.** For TTS, SpeechOp (Figure 3, right) processes a text transcript. We extract transcript representations with a frozen, pre-trained ByT5-base encoder Xue et al. (2022); Lovelace et al. (2024a). ByT5's character-level representations capture phonetic information crucial for natural speech. The DiT is conditioned on the ByT5 embeddings via cross-attention, dynamically aligning text and audio frames and guiding denoising based on text content. For our Diffusion

Transformer (DiT) architecture, we incorporate design choices from recent TTS systems (Lee et al., 2024; Lovelace et al., 2024b) (full details in the Appendix).

To enable speaker-prompted generation and speech editing, we train our model to perform inpainting for 75% of samples Le et al. (2024). After adding noise from the forward diffusion process, we replace a random segment of the latent with the clean, target segment. We additionally sum a learnable binary embedding at the input layer to distinguish clean from noisy frames. The network will then learn to extrapolate speaker and speech properties from the ground-truth region to denoise the noisy speech. For half of our inpainting samples, we replace the initial segment (simulating voice prompts). In the other half, we noise only the middle section, replacing the start and end of the utterance with clean speech to simulate speech editing. For sampling the relative duration, we follow Lovelace et al. (2024b) and use a Beta distribution with a mode of .01 and a concentration of 5 to emphasize challenging cases with short prompts.

**Speech-to-Speech Pathway.** To handle speech-to-speech tasks like enhancement and separation (Figure 3, left), SpeechOp introduces a dedicated Audio Encoder to process source audio such as noisy speech. This encoder adopts the same DiT architecture as the main model but starts with random initialization. Since speech-to-speech tasks inherently maintain frame-level alignment between source and target audio, we implement a straightforward frame-wise mixing approach rather than use a complex alignment mechanism. Specifically, the Audio Encoder's output representations are directly added to the Diffusion Latent before processing by the Diffusion Transformer, allowing direct incorporation of source audio information during denoising. To handle different speech-to-speech tasks, we use a learnable Task Embedding that conditions both the Audio Encoder and Diffusion Transformer. Following the class embedding approach from the original DiT paper (Peebles & Xie, 2023), the Diffusion Transformer sums the task embedding with the timestep embedding to condition the network via adaptive normalization (AdaLN), while the Audio Encoder uses the task embedding directly for adaptive normalization. This simple approach provides effective task-specific guidance to both components based on the desired operation (enhancement, separation, etc.).

Some speech-to-speech tasks require additional input prompts. For example, speaker separation needs a reference speech sample to identify the target speaker, while acoustic matching needs an example of the target acoustics. In these cases, we prepend the prompt to both the source audio and noisy latent to maintain frame-wise alignment. For tasks that typically don't use prompts (like enhancement), we unmask the latent's initial segment in 10% of training instances to enable transfer learning with speaker-prompted TTS. These prompt durations follow the same Beta distribution used for TTS inpainting.

**Multi-Task Fine-Tuning.** SpeechOp uses a two-stage training approach. After initial TTS pre-training, we conduct multi-task fine-tuning where both the Audio Encoder and pre-trained DiT backbone are jointly optimized. During this stage, we sample TTS and speech-to-speech (S2S) data with equal frequency. Within the S2S samples, we apply selective upsampling - tripling the frequency of enhancement and speaker separation examples since these are the most challenging tasks. This two-stage strategy efficiently adapts the TTS model into our multi-task SpeechOp model.

**Diffusion Training.** During training, we sample noise levels using a shifted cosine schedule (s=0.5) (Hoogeboom et al., 2023), following Lovelace et al. (2024a). We employ the Sigmoid diffusion loss weighting from Hoogeboom et al. (2024) with a bias of -2.5 to concentrate training on perceptually relevant noise levels. To enable classifier-free guidance during inference, we randomly drop conditioning information (source audio and transcript) 10% of the time during training (Ho & Salimans, 2022).

# 6 INFERENCE-TIME TASK COMPOSITION

The ability to compose speech operations—such as simultaneously enhancing noisy speech while restoring its content via text—represents a powerful capability for speech processing. Text-guided generation can help produce a plausible, high-fidelity version of content that is otherwise not recoverable from complex acoustic situations, such as intense noise and reverberation encountered in speech enhancement. Similarly, in speaker separation, the text of spoken content could provide important contextual cues for disentangling speakers. Nonetheless, achieving an effective composition of tasks poses significant technical challenges.

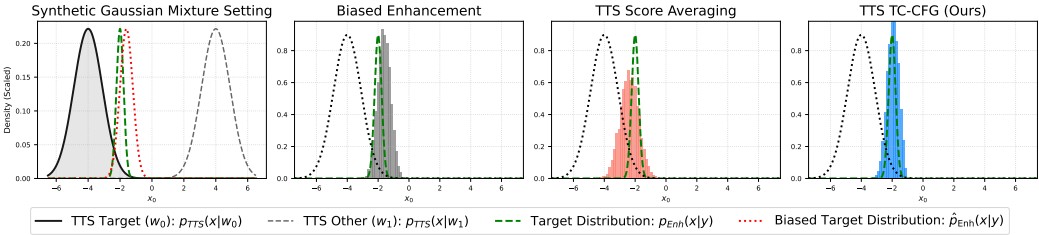

Figure 4: **A 1D toy simulation illustrating different task composition methods. (a)** The setup, showing a bimodal TTS prior (target $w_0$ and other $w_1$), an ideal sharp target distribution ($p_{\text{Enh}}$), and a biased enhancement model ($\hat{p}_{\text{Enh}}$) whose output is misaligned. **(b)** Samples from the unguided biased model. **(c)** Samples using score averaging (Eq. equation 1). **(d)** Samples using our TC-CFG method (Eq. equation 4).

Prior work, including Fugatto in the audio domain, typically computes a weighted average of score functions to compose operationsLiu et al. (2022); Valle et al. (2025), like for enhancement and TTS:

$$\mathbf{s}_\theta^{\text{avg}}(\mathbf{z}_t|y, w) = (1 - \alpha)\mathbf{s}_\theta^{\text{enh}}(\mathbf{z}_t|y) + \alpha\mathbf{s}_\theta^{\text{tts-prior}}(\mathbf{z}_t|w) \tag{1}$$

Here, $\mathbf{s}_\theta^{\text{enh}}(\mathbf{z}_t|y)$ is the score from an enhancement model conditioned on noisy audio $y$, and $\mathbf{s}_\theta^{\text{tts}}(\mathbf{z}_t|w)$ represents a score function derived from a TTS model aiming to generate speech for transcript $w$. While straightforward, this approach poses a fundamental limitation: it combines the generative priors of different tasks. For speech enhancement with TTS guidance, direct averaging allows the TTS model's broad acoustic prior (learned from diverse data for generation) to corrupt the enhancement model's focused studio-quality prior (learned for reconstruction), degrading output quality.

To address this challenge, we propose decomposing the desired score function $\nabla_{\mathbf{z}_t} \log p(\mathbf{z}_t|y, w)$ into task-specific components. Using Bayes' rule and a conditional independence assumption (transcript $w$ is independent of noisy audio $y$ given latent $\mathbf{z}_t$; detailed derivation in the Appendix), we arrive at:

$$\nabla_{\mathbf{z}_t} \log p(\mathbf{z}_t|y, w) = \nabla_{\mathbf{z}_t} \log p(\mathbf{z}_t|y) + \nabla_{\mathbf{z}_t} \log p(w|\mathbf{z}_t). \tag{2}$$

This decomposition yields two complementary terms: an enhancement score $\nabla_{\mathbf{z}_t} \log p(\mathbf{z}_t|y)$ that guides acoustic quality based on the input $y$, and a discriminative guide $\nabla_{\mathbf{z}_t} \log p(w|\mathbf{z}_t)$. This second term leverages a TTS model not for its generative prior, but for its ability to *discriminate* whether a latent $\mathbf{z}_t$ is likely to produce content matching transcript $w$. This term guides the latent towards speech aligned with the transcript without imposing the TTS model's full acoustic prior.

**Implementation via Classifier-Free Guidance.** We implement this decomposition using classifier-free guidance (CFG) Ho & Salimans (2022) to approximate the discriminative signal $\nabla_{\mathbf{z}_t} \log p(w|\mathbf{z}_t)$:

$$\nabla_{\mathbf{z}_t} \log p(w|\mathbf{z}_t) \approx \gamma \left(\mathbf{s}_\theta^{\text{tts}}(\mathbf{z}_t|w) - \mathbf{s}_\theta^{\text{tts}}(\mathbf{z}_t)\right) \tag{3}$$

where $\mathbf{s}_\theta^{\text{tts}}(\mathbf{z}_t|w)$ is the score of a TTS model conditioned on transcript $w$, $\mathbf{s}_\theta^{\text{tts}}(\mathbf{z}_t)$ is its unconditional score, and $\gamma$ is a guidance scale. Substituting this into Eq. equation 2, our final composed score is:

$$\mathbf{s}_\theta^{\text{CFG}}(\mathbf{z}_t|y, w) \approx \mathbf{s}_\theta^{\text{enh}}(\mathbf{z}_t|y) + \gamma \left(\mathbf{s}_\theta^{\text{tts}}(\mathbf{z}_t|w) - \mathbf{s}_\theta^{\text{tts}}(\mathbf{z}_t)\right). \tag{4}$$

This formulation, which we term Task-Composition Classifier-Free Guidance (TC-CFG), preserves the strengths of both tasks. The enhancement term maintains acoustic quality and speaker characteristics. The CFG-derived discriminative term provides content alignment by isolating text-specific guidance, avoiding the pitfalls of directly mixing generative TTS priors with the enhancement prior.

**Synthetic Simulation.** To illustrate the behavior of score averaging and motivate our TC-CFG approach, we present results for a 1D Gaussian mixture simulation (Figure 4, full details in appendix), where the score functions are analytically tractable. We present this example primarily to provide intuition for the behavior of the two approaches. We empirically validate the benefits of our approach for real speech processing applications in section 8.

Our setup (Figure 4a) features a bimodal TTS prior, a sharp ideal enhanced distribution $p_{\text{Enh}}$ for the target word $w_0$, and an imperfect (biased) enhancement model $\hat{p}_{\text{Enh}}$ whose content $w_0$ is misaligned.

Without guidance, the biased model's samples are incorrect (Figure 4b). However, combining the biased enhancement score function with the TTS score function can potentially correct for content errors. Score averaging (Figure 4c) pulls samples towards the $w_0$ TTS mode. However, because this mixes in the broad TTS prior, the result is a "smeared" distribution that deviates from the enhancement distribution. In contrast, our TC-CFG approach (Figure 4d) incorporates discriminative TTS guidance (via $\nabla_{x_t} \log p_{\text{TTS}}(w_0|x_t)$) to steer sampling. This shifts the sampled distribution to satisfy the discriminative signal without compromising the enhancement prior.

## 7 EXPERIMENTAL SETUP

SpeechOp integrates a 20-layer Diffusion Transformer (DiT, 419M parameters) with an 8-layer audio encoder (71M parameters). We compare it against strong baselines for speech enhancement and speaker separation.

**Training Data.** For TTS, we combine MLS English ( 44k hours) (Pratap et al., 2020) for longer utterances (10-20s) and Libri-TTS (585 hours) (Zen et al., 2019) for shorter segments (<10s), improving robustness. All audio is resampled to 48kHz and transcripts lowercased. For S2S tasks, we use LibriTTS-R (Koizumi et al., 2023a) for clean speech and simulate degradations using established noise/impulse response datasets and pipelines (Yang et al., 2024), creating 5s paired instances. Further dataset details are in the Appendix.

**Tasks and Baselines. Text-to-Speech (TTS):** We evaluate on LibriSpeech test-clean (Panayotov et al., 2015) against contemporary end-to-end TTS systems (Le et al., 2024; Chen et al., 2024b; Lee et al., 2024). Speech editing is evaluated on the LibriTTS portion of RealEdit (Peng et al., 2024). **Speech Enhancement (SE):** Baselines include waveform (StoRm (Lemercier et al., 2023)) and diffusion-based (SGMSE+ (Richter et al., 2023), Miipher+WavLM (Koizumi et al., 2023b; Yang et al., 2024)) models, and GAN-based HiFi-GAN-2 (Su et al., 2021a). **Speaker Separation (SS):** We compare against SepFormer variants (Subakan et al., 2021; Chen et al., 2024a), including those trained on WHAMR! (Maciejewski et al., 2020) and with acoustic-content simulation.

**Evaluation Metrics.** We assess SpeechOp on four dimensions: **Subjective Quality:** Mean Opinion Scores (MOS, 1-5 scale) from listening tests on Prolific (pro) (details in Appendix A). **Signal Similarity:** PESQ (perceived quality), MCD (spectral distance, lower is better), and SI-SDRi (separation distortion improvement) Roux et al. (2019). **Neural Similarity:** WavLM-TDCNN (Chen et al., 2021b) for speaker similarity (SIM) and SpeechBERTScore (SpBS) Saeki et al. (2024) for semantic alignment. **Content Accuracy:** Word Error Rate (WER) via HuBERT-L Hsu et al. (2021) for TTS and WhisperX (large-v2) Radford et al. (2023); Bain et al. (2023) for other tasks.

## 8 RESULTS AND DISCUSSION

**Text-To-Speech.** To examine the impact of multi-task training on text-to-speech, we evaluate our model's zero-shot TTS performance with 3 second speech prompts. Crucially, we initialize SpeechOp from our TTS Baseline, allowing us to directly assess the impact of multi-task training. Table 1 demonstrates that SpeechOp not only preserves but *enhances* zero-shot TTS capabilities. After undergoing multi-task training, SpeechOp improves performance across all MOS metrics and objective speaker similarity compared to the TTS Baseline, with minimal loss of intelligibility. Exposure to tasks like enhancement and separation likely enhances SpeechOp's ability to generalize and generate natural speech across diverse acoustic environments.

Against recent TTS systems of comparable scale, SpeechOp exhibits strong performance, matching or exceeding CLaM-TTS and XTTS on most metrics. Impressively, it also surpasses the larger VoiceCraft model in intelligibility and subjective quality. While DiTTo-TTS, a larger model trained on more diverse data, achieves higher overall scores, SpeechOp's results are highly competitive within its class. Future work will explore scaling SpeechOp to leverage similar large-scale datasets.

Beyond zero-shot TTS, SpeechOp demonstrates state-of-the-art capabilities in speech editing. As shown in Table 2, SpeechOp significantly outperforms VoiceCraft across all subjective MOS metrics, despite having fewer parameters. These results validate the robustness of our multi-task approach, as SpeechOp maintains exceptional speech editing performance while supporting multiple tasks.

Table 1: Zero-Shot Text-to-Speech Evaluation. MOS metrics evaluate different aspects: MOS-Q (Quality), MOS-N (Naturalness), MOS-VS (Voice Similarity), and MOS-SS (Style Similarity). Models in a different parameter regime are displayed in gray.

| Model | Params | Training Data | WER ↓ | SIM ↑ | MOS-Q ↑ | MOS-N ↑ | MOS-VS ↑ | MOS-SS ↑ |
|---|---|---|---|---|---|---|---|---|
| Ground Truth | — | — | 2.19 | 0.67 | $4.24_{\pm 0.06}$ | $4.16_{\pm 0.06}$ | $3.79_{\pm 0.06}$ | $3.60_{\pm 0.06}$ |
| DiTTo-TTS (Lee et al., 2024) | 740M | ∼56k hrs | 2.56 | .62 | $4.16_{\pm 0.04}$ | $4.14_{\pm 0.04}$ | $4.17_{\pm 0.04}$ | $4.02_{\pm 0.04}$ |
| VoiceCraft (Peng et al., 2024) | 830M | ∼69k hrs | 6.32 | .61 | $3.66_{\pm 0.04}$ | $3.65_{\pm 0.05}$ | $3.43_{\pm 0.05}$ | $3.38_{\pm 0.05}$ |
| CLaM-TTS (Kim et al., 2024) | 584M | ∼56k hrs | 5.11 | .49 | $3.67_{\pm 0.04}$ | $3.70_{\pm 0.04}$ | $3.69_{\pm 0.05}$ | $3.54_{\pm 0.05}$ |
| XTTS (Casanova et al., 2024) | 482M | ∼17k hrs | 4.93 | .49 | $3.76_{\pm 0.04}$ | $3.66_{\pm 0.05}$ | $3.28_{\pm 0.05}$ | $3.27_{\pm 0.05}$ |
| TTS Baseline (Ours) | 419M | ∼45k hrs | 3.32 | .48 | $3.65_{\pm 0.05}$ | $3.56_{\pm 0.05}$ | $3.31_{\pm 0.05}$ | $3.25_{\pm 0.05}$ |
| SpeechOp (Ours) | 419M | ∼45k hrs | 3.57 | .53 | $3.86_{\pm 0.04}$ | $3.69_{\pm 0.05}$ | $3.67_{\pm 0.05}$ | $3.58_{\pm 0.05}$ |
| Δ from Multi-Task Training | — | — | +0.25 | +.05 | $\mathbf{+0.22}_{\pm 0.06}$ | $\mathbf{+0.13}_{\pm 0.07}$ | $\mathbf{+0.36}_{\pm 0.07}$ | $\mathbf{+0.32}_{\pm 0.07}$ |

Table 2: Speech Editing Evaluation.

| Model | Params | Training Data | WER ↓ | MOS-Q ↑ | MOS-N ↑ | MOS-VS ↑ | MOS-SS ↑ |
|---|---|---|---|---|---|---|---|
| Ground Truth | — | — | 16.2 | $4.33_{\pm 0.04}$ | $4.40_{\pm 0.03}$ | $4.66_{\pm 0.03}$ | $4.63_{\pm 0.03}$ |
| VoiceCraft (Peng et al., 2024) | 830M | ∼69k hrs | 16.3 | $3.62_{\pm 0.04}$ | $3.99_{\pm 0.04}$ | $4.12_{\pm 0.04}$ | $4.01_{\pm 0.04}$ |
| TTS Baseline (Ours) | 419M | ∼45k hrs | 16.4 | $4.18_{\pm 0.04}$ | $4.23_{\pm 0.04}$ | $4.45_{\pm 0.03}$ | $4.23_{\pm 0.04}$ |
| SpeechOp (Ours) | 419M | ∼45k hrs | 15.9 | $4.15_{\pm 0.04}$ | $4.19_{\pm 0.04}$ | $4.48_{\pm 0.03}$ | $4.25_{\pm 0.03}$ |

**Speech Enhancement.** Our Implicit Task Composition (ITC) pipeline integrates ASR transcripts from Whisper with our TC-CFG method (Section 6) to guide speech content. We find that our ITC pipeline achieves state-of-the-art content preservation in speech enhancement. As shown in Table 3, ITC yields a Word Error Rate (WER) of 2.9%, a 46% relative reduction over the strong HiFi-GAN-2 baseline, significantly reducing the content loss common with generative models. Our ITC pipeline leverages web-scale knowledge from ASR models without requiring transcriptions for training the enhancement component itself.

Our ITC's transcript guidance method is more flexible than transcript-conditioned S2S models. Such models can struggle when ASR errors create contradictions between acoustic and textual inputs, or their performance may be upper-bounded by the input audio quality if they cannot generatively restore highly corrupted content. Furthermore, they typically lack control over the influence of the transcript versus the acoustics at inference time. In contrast, TC-CFG (Eq. equation 4) provides this control through a tunable guidance strength ($\gamma$). This allows SpeechOp to trade-off prioritizing acoustic fidelity or emphasizing content restoration guided by the transcript depending on the application.

Table 3: Speech Enhancement Results. (**Left**) Quantitative metrics. (**Right**) Subjective MOS scores with standard error.

| Model | PESQ ↑ | MCD ↓ | SpBS ↑ | WER ↓ |
|---|---|---|---|---|
| Noisy Source Audio | 1.12 | 11.22 | .888 | 3.3 |
| StoRm | 1.61 | 6.36 | .883 | 7.0 |
| Miipher | 1.44 | 5.15 | .898 | 7.0 |
| SGMSE+ | 1.98 | 5.28 | .923 | 5.7 |
| HiFi-GAN-2 | 2.23 | 4.40 | .934 | 5.4 |
| SpeechOp (No Transcript) | 2.00 | 4.83 | .908 | 8.1 |
| +ITC | 2.05 (+0.05) | 4.85 (+0.02) | .928 (+.020) | 2.9 (**-5.2**) |
| +Speaker Personalization | 2.12 (+0.07) | 4.69 (**-0.16**) | .926 (-.002) | 2.4 (**-0.5**) |
| SpeechOp (Gold Transcript) | 2.06 | 4.83 | .931 | 2.1 |

| Model | MOS ↑ |
|---|---|
| Noisy Source Audio | $1.78_{\pm 0.07}$ |
| SGMSE+ | $3.76_{\pm 0.03}$ |
| HiFi-GAN-2 | $3.90_{\pm 0.04}$ |
| SpeechOp (No Transcript) | $3.93_{\pm 0.04}$ |
| SpeechOp-ITC (WhisperX) | $3.89_{\pm 0.04}$ |
| Clean Reference Audio | $4.67_{\pm 0.02}$ |

Even using Whisper transcripts derived from the noisy source audio, ITC improves content intelligibility over the original audio (WER 2.9% vs. 3.3%) and enhancement without transcripts (WER 8.1%). This suggests TC-CFG effectively balances the acoustic information from the noisy source audio with the imperfect guidance from ASR transcription. While signal-fidelity metrics often penalize generative outputs, SpeechOp's ITC matches HiFi-GAN-2's subjective quality (Table 3 Right) while delivering superior content accuracy.

---

[1]We compare SepFormer models in Chen et al. (2024a) since they support speaker separation in multiple acoustic environments.

Table 4: Speaker Separation Evaluation (Subj.). We report the average MOS and the standard error.

| Model | LibriMix Clean | LibriMix Noise | WHAMR | WSJ0-2Mix | Total |
|---|---|---|---|---|---|
| SepFormer Chen et al. (2024a) | $3.32 \pm 0.07$ | $2.95 \pm 0.07$ | $3.06 \pm 0.07$ | $3.53 \pm 0.07$ | $3.22 \pm 0.04$ |
| DM SepFormer Chen et al. (2024a) | $3.59 \pm 0.07$ | $2.67 \pm 0.07$ | $2.53 \pm 0.07$ | $3.58 \pm 0.07$ | $3.10 \pm 0.04$ |
| AC-SIM SepFormer Chen et al. (2024a) | $3.74 \pm 0.07$ | $2.81 \pm 0.07$ | $2.53 \pm 0.07$ | $3.65 \pm 0.07$ | $3.20 \pm 0.04$ |
| AC-SIM-ML SepFormer Chen et al. (2024a) | $3.74 \pm 0.06$ | $3.02 \pm 0.07$ | $2.64 \pm 0.07$ | $3.66 \pm 0.06$ | $3.28 \pm 0.04$ |
| SpeechOp (No Transcript) | $3.86 \pm 0.07$ | $3.68 \pm 0.07$ | $2.89 \pm 0.08$ | $3.77 \pm 0.07$ | $3.57 \pm 0.04$ |
| SpeechOp (Gold Transcript) | $4.13 \pm 0.06$ | $4.21 \pm 0.06$ | $3.37 \pm 0.08$ | $3.91 \pm 0.06$ | $3.92 \pm 0.03$ |
| Mixture | $1.38 \pm 0.05$ | $1.35 \pm 0.04$ | $1.39 \pm 0.04$ | $1.33 \pm 0.05$ | $1.36 \pm 0.02$ |
| Clean Target | $4.26 \pm 0.06$ | $4.48 \pm 0.05$ | $4.29 \pm 0.06$ | $4.00 \pm 0.06$ | $4.25 \pm 0.03$ |

Table 5: Quantitative Speaker Separation Performance on the WSJ0-2Mix Dataset.

| Method | SI-SDRi ↑ | MCD ↓ | SpBS ↑ | WER ↓ |
|---|---|---|---|---|
| SepFormer [1] Chen et al. (2024a) | 11.86 | 1.72 | .929 | 4.4 |
| AC-SIM-ML SepFormer Chen et al. (2024a) | 11.80 | 1.55 | .931 | 6.8 |
| SpeechOp (No Transcript) | 0.23 | 4.11 | .899 | 11.1 |
| SpeechOp (Gold Transcript) | 0.53 | 4.20 | .919 | 5.5 |

SpeechOp also enables novel applications like personalized enhancement by composing enhancement with voice cloning. Given a clean voice sample from the target speaker, we compose the enhancement task with speaker-prompted TTS via TC-CFG, allowing the model to simultaneously enhance audio quality while adhering to the target speaker's voice characteristics. This composition improves speaker fidelity (MCD, PESQ) and modestly reduces WER. To provide an upper bound, ground-truth transcripts lead to a 2.1% WER. Across all scenarios, SpeechOp's ITC, with our composition approach, effectively integrates textual guidance for controllable, content-aware speech enhancement.

**Speaker Separation.** On human Mean Opinion Score (MOS)—the gold-standard metric for perceived speech quality—SpeechOp *significantly outperforms* SepFormer baselines across all datasets (Table 4). Despite these human-rated gains, SpeechOp attains lower objective signal-fidelity metrics (e.g., SI-SDRi, MCD on WSJ0-2Mix; Table 5), reflecting a known mismatch between signal-level metrics and perceived quality for generative models Erdogan et al. (2023); Chen et al. (2024a). This divergence stems from methodology: tra-

Table 6: **Task Composition.** We compare our proposed composition formulation (TC-CFG) against averaging the score vectors (TC-Avg). Gold transcripts are used in this ablation.

| Model | PESQ ↑ | MCD ↓ | SpBS ↑ | WER ↓ |
|---|---|---|---|---|
| Noisy Source Audio | 1.12 | 11.22 | .888 | 3.3 |
| SpeechOp (No Transcript) | 2.00 | 4.83 | .908 | 8.1 |
| SpeechOp (TC-Avg) | 1.88 | 5.24 | .909 | 3.4 |
| SpeechOp (TC-CFG) (Ours) | 2.06 | 4.83 | .931 | 2.1 |
| $\Delta$ (TC-CFG vs TC-Avg) | **+.18** | **-0.42** | **+.022** | **-1.3** |

ditional mask-based separators optimize signal reconstruction, whereas our generative approach prioritizes naturalness and perceptual quality rather than strict mixture consistency. Importantly, transcript guidance markedly improves content preservation, reducing WER from 11.1% to 5.5% with ground-truth transcripts, showing that SpeechOp can leverage textual information to boost separation accuracy while maintaining its perceptual strengths. We note that our current evaluation uses fully overlapping synthetic mixtures following standard benchmarks (LibriMix, WSJ0-2Mix). Extending SpeechOp to realistic conversational scenarios with partial overlap, combined with diarization-assisted ASR to provide speaker-attributed transcripts, is a promising direction for future work.

**Task Composition Ablation.** We empirically validate our TC-CFG approach by composing SpeechOp's enhancement capability with TTS-based textual guidance from the gold transcripts (Table 6). The "SpeechOp (No Transcript)" baseline represents the performance of our enhancement model without any textual guidance. When employing the score averaging approach ("SpeechOp (TC-Avg)"), we observe a degradation in signal fidelity metrics compared to the "No Transcript" baseline (e.g. MCD increases from 4.83 to 5.24). This aligns with the intuition from our synthetic simulation (Figure 4c), where averaging with the broader TTS prior can negatively impact the focused prior of

the enhancement model. While TC-Avg does improve content preservation (WER 3.4% vs. 8.1% for "No Transcript"), this comes at the cost of acoustic quality and signal fidelity.

In contrast, our proposed composition approach, TC-CFG, demonstrates superior performance across all metrics. It not only achieves the best content preservation with a WER of 2.1% (a 38% reduction over TC-Avg's 3.4% WER), but it also *maintains or improves* signal fidelity compared to the "No Transcript" baseline (e.g. PESQ 2.06 vs. 2.00). These results empirically confirm that our TC-CFG formulation effectively isolates text-conditional guidance without degrading acoustic quality. This allows SpeechOp to leverage knowledge from the TTS model for robust content preservation (low WER) while simultaneously maintaining, and even slightly enhancing, the acoustic quality and speaker characteristics established by the enhancement model. This careful decomposition of task-specific guidance is crucial for enabling effective and high-fidelity task composition in generative speech processing.

## 9 FUTURE WORK

Several promising directions emerge from this work. TC-CFG's principled decomposition of score functions naturally extends beyond the speech enhancement and separation tasks explored here. For instance, voice conversion could leverage the discriminative signal from a speaker-conditioned model to guide a content-preserving base model, and extending the framework to additional tasks (e.g., text-to-audio generation) would further expand the space of composable operations. More broadly, TC-CFG generalizes to any setting where multiple generative tasks can be combined at inference time, including modalities beyond speech. Within speech processing, conditioning on self-supervised learning (SSL) features could improve the baseline intelligibility of enhancement or separation models (Guimarães et al., 2025), and TC-CFG composition can similarly be applied on top of such an SSL-conditioned model. Further exploration of task conditioning and specialization strategies could also yield improvements in multi-task performance. Finally, extending SpeechOp to cross-lingual settings represents an important direction for building truly universal speech processing systems.

## 10 CONCLUSION

In this work, we addressed a fundamental data disparity between text-to-speech synthesis and speech-to-speech tasks by adapting pre-trained TTS models to enable high-quality speech processing despite limited paired data. Through SpeechOp, we showed that multi-task training not only enables flexible speech-to-speech processing but also improves the underlying TTS quality, demonstrating that these tasks are mutually beneficial rather than competing objectives. Our Task-Composition Classifier-Free Guidance (TC-CFG) provides a principled mechanism for composing speech operations at inference time, using the TTS model as a discriminative guide rather than mixing generative priors. Building on this, our Implicit Task Composition (ITC) framework demonstrated how to leverage web-scale speech understanding from discriminative ASR models to achieve state-of-the-art content preservation without requiring parallel transcript data during training. By bridging the gap between data-rich and data-constrained speech tasks, this work opens new possibilities for unified, scalable speech processing systems.

## 11 ETHICS STATEMENT

Our work advances controllable, generative speech reconstruction for beneficial applications such as accessibility (clearer listening and captioning), restoration of degraded or archival audio, personalized but consented enhancement, and robust low-bandwidth communication.

We recognize the potential for misuse of such generative technology including impersonation/deepfakes. To mitigate these risks, we restrict experiments to publicly available datasets and we recommend deployment guardrails such as watermarking (O'Reilly et al., 2024) when releasing models. For human studies, raters were consenting adults performing non-sensitive listening/MOS tasks and were compensated at fair market rates

## 12 REPRODUCIBILITY STATEMENT

Our models are trained and evaluated on publicly available data. We provide a complete description regarding the datasets, model architecture, and evaluations to enable faithful reproduction of this work.

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

## A    SUBJECTIVE STUDY DETAILS

**Methodology.** Our studies used native English speakers on Prolific to measure naturalness, quality, voice similarity, and style similarity on a 1-5 scale, for text-to-speech synthesis based on a reference speech sample. For our speech processing tasks, we measure the quality of the audio sample. We also included a flag for unintelligible content, though no samples were ultimately flagged by a majority of raters.

**Quality Control.** To filter out unreliable ratings for the TTS and speech editing studies, we used two types of hidden validation tests. The first was a mismatched speaker test (different but real speakers for reference and sample); if a participant rated speaker similarity > 3, their ratings were discarded. The second was an identical pair test; if any attribute was rated < 4, their ratings were discarded. For the speech processing tasks, we conducted similar validation tests with the clean and noisy audio samples.

**Participant details.** For all of our subjective tests, each worker rated 30 samples, including 4 validation tests. Our TTS study involved 288 unique workers rating 80 utterances per method. Our speech editing study involved 151 unique workers rating 100 utterances per method. Our enhancement study involved 236 unique workers rating 96 utterances per method.

**Compensation.** For our listening experiments, participants were compensated at a rate of $15/hour, which is above the platform's recommendation.

## B    SIMULATION STUDY IMPLEMENTATION DETAILS

For our guidance comparison simulation study on the 1D Gaussian Mixture Model, we provide detailed implementation specifics to ensure reproducibility.

The 1D GMM setting enables exact computation of all relevant quantities, providing a controlled environment for comparing guidance strategies. Both the conditional score function, $\nabla_{x_t} \log p_t(x_t|y)$, and guidance term, $\nabla_{x_t} \log p_{\text{TTS}}(y|x_t)$, can be computed analytically.

Our synthetic experiments use the configuration detailed in Table 7.

### B.1    GUIDANCE STRATEGY COMPARISON

Our simulation compares three fundamental approaches to combining TTS and speech enhancement models:

**No Guidance:** Uses only the imperfect enhancement model's score function, representing current single-task approaches:
$$s_{\text{total}} = s_{\text{enh}}(x_t, \sigma_t)$$

**CFG-Style Guidance:** Augments the enhancement model with discriminative guidance from the TTS model using classifier-free guidance:
$$s_{\text{total}} = s_{\text{enh}}(x_t, \sigma_t) + \rho \cdot \nabla_{x_t} \log p_{\text{TTS}}(y|x_t)$$

where the guidance term $\nabla_{x_t} \log p_{\text{TTS}}(y|x_t)$ leverages the TTS model's ability to distinguish content-matching samples.

**Score Averaging:** Linearly combines the enhancement model score with the true conditional TTS score:
$$s_{\text{total}} = (1 - \alpha) \cdot s_{\text{enh}}(x_t, \sigma_t) + \alpha \cdot s_{\text{TTS}}(x_t, \sigma_t|y)$$

This approach directly mixes the score functions from both models.

### B.2    NOISE SCHEDULE AND SAMPLING

We employ a log-linear interpolation for noise levels: $\sigma_t = \exp\left(\frac{t}{T} \log(\sigma_{\text{final}}) + \frac{T-t}{T} \log(\sigma_{\text{init}})\right)$

The update step follows the variance exploding diffusion formulation Karras et al. (2022): $x_{t+1} = x_t + (\sigma_t^2 - \sigma_{t+1}^2) \cdot s_{\text{total}} + \sqrt{\sigma_t^2 - \sigma_{t+1}^2} \cdot \epsilon$ where $\epsilon \sim \mathcal{N}(0, 1)$.

Table 7: Guidance Comparison Simulation Parameters

| Parameter | Value |
|---|---|
| **Base Parameters** | |
| *TTS Distribution (Generic Speech)* | |
| Component means | $\mu_0 = -4.0, \mu_1 = 4.0$ |
| Component std. devs. | $\sigma_0 = \sigma_1 = 0.9$ |
| Component weights | $w_0 = w_1 = 0.5$ |
| Target component | $y = 0$ |
| *Enhancement Transforms* | |
| Mean shift | $\Delta\mu = 2.0$ |
| Variance reduction factor | $\gamma = 4$ |
| Imperfect model bias | $\epsilon = 0.4$ |
| Imperfect model variance inflation | $\beta = 1.8$ |
| **Derived Parameters** | |
| *True Enhanced Speech* | |
| Mean | $\mu_0 + \Delta\mu = -2.0$ |
| Std. dev. | $\sigma_0/\gamma = 0.23$ |
| *Imperfect Enhancement Model* | |
| Mean | $\mu_0 + \Delta\mu + \epsilon = -1.6$ |
| Std. dev. | $\beta \cdot \sigma_0/\gamma = 0.41$ |
| *Sampling Parameters* | |
| Number of samples | 5000 |
| Number of timesteps | 200 |
| Initial noise level | $\sigma_{\max} = 80$ |
| Final noise level | $\sigma_{\min} = 0.005$ |
| *Guidance Parameters* | |
| CFG guidance strength | $\rho = 10^4$ |
| Score averaging weight | $\alpha = 0.5$ |

### B.3 EVALUATION METRICS

We evaluate the final samples using KL divergence computed between the empirical distribution of generated samples and the true enhanced speech distribution, representing the ideal outcome.

## C AUDIO AUTOENCODER

For efficient latent diffusion modeling, we develop an autoencder based on DAC (Kumar et al., 2023) but with a continuous variational bottleneck instead of residual vector quantization. For 48 kHz input audio $\mathbf{y} \in \mathbb{R}^{1 \times T}$, the encoder $E$ maps to latent representations $\mathbf{x}_0 = E(\mathbf{y})$ with dimensions $\mathbb{R}^{C \times L}$, where $C = 64$ is the latent channel dimension and $L$ is the temporal dimension downsampled by a factor of 1200 (resulting in a 40 Hz latent representation). The decoder $D$ mirrors this architecture to reconstruct the waveform.

The encoder's output is transformed into latent variables through a variational bottleneck that models the approximate posterior $q(\mathbf{z}|\mathbf{y})$:

$$\mathbf{z} = \boldsymbol{\mu} + \boldsymbol{\sigma} \odot \boldsymbol{\epsilon}, \quad \text{where} \quad \boldsymbol{\epsilon} \sim \mathcal{N}(0, \mathbf{I}) \tag{5}$$

The model is trained to minimize reconstruction loss and KL divergence:

$$\mathcal{L}_{\text{AE}} = \mathbb{E}_{\mathbf{y}}[\|\mathbf{y} - \hat{\mathbf{y}}\|_1] + \lambda_{\text{KL}}\mathcal{L}_{\text{KL}} \tag{6}$$

where $\lambda_{\text{KL}} = 0.1$ balances the objectives. We also employ adversarial training with a complex STFT discriminator following DAC to improve reconstruction quality.

## D    ACOUSTIC SIMULATION

First, we randomly select a clean speech sample and apply random equalization and compression. Background noise is then added at a signal-to-noise ratio (SNR) of $-10$ to 30 dB. We randomly apply reverberation using impulse response (IR) samples. Additional degradation like random bandlimiting down to 1kHz is applied to simulate input at various sample rates. We dynamically generate training pairs during training to increase diversity of the degradation combinations.

We trained the models with public datasets at 44.1k sample rate. The clean speech data is sourced from LibriTTS-R (Koizumi et al., 2023a) and upsampled to 44.1k sample rate via bandwidth extension (Su et al., 2021b). The noise samples include the DNS Challenge (Dubey et al., 2024) and SFS-Static-Dataset (Chen et al., 2022). The impulse response (IR) data includes MIT IR Survey (Traer & McDermott, 2016), EchoThief (ech), and OpenSLR28 (Ko et al., 2017).

## E    ARCHITECTURE AND TRAINING DETAILS

### E.1    MODEL ARCHITECTURE

We present our model architecture details in Table 8. For our transfer learning experiments and architecture ablation, we utilize a smaller version of SpeechOp with 12 DiT layers and 6 encoder layers. We also incorporate dense connections (Lee et al., 2024), a position-aware cross-attention mechanism ((Lovelace et al., 2024b)), and append 8 register tokens to process global information (Lovelace et al., 2024b). We condition on the learnable task embedding by summing it with the timestep embedding that is given to the DiT network.

Table 8: **SpeechOp Architecture Parameters**

| Parameter | Value |
| --- | :---: |
| *Diffusion Transformer* | |
| Audio Latent Dimension | 64 |
| Model Dimension | 1024 |
| Feed-forward Dimension | 3072 |
| Attention Heads | 8 |
| Number of Layers | 20 |
| Dropout | 0.1 |
| *Audio Encoder* | |
| Model Dimension | 768 |
| Feed-forward Dimension | 2304 |
| Number of Layers | 8 |
| *Common Components* | |
| Position Encoding | Rotary |
| Layer Normalization | AdaLN ($\epsilon$=1e-5) |
| Activation | SwiGLU |
| Text Encoder | ByT5-base |

### E.2    TRAINING CONFIGURATION

All model training is distributed across 32 Nvidia A100s. Training proceeds in two stages:

**Stage 1: TTS Pre-training**    Model is trained for 400K iterations with a batch size of 4 per GPU. We use AdamW optimization with learning rate 2e-4 and weight decay 0.1. Training employs 4000 warmup steps and we perform two steps of gradient accumulation.

**Stage 2: Multi-task Fine-tuning**    Starting from the pre-trained TTS model, we extend the encoder to 8 layers and train for an additional 200K iterations. We use a lower learning rate of 1e-4 and

weight decay of 0.01, with two steps of gradient accumulation. Batch sizes are 4 for TTS and 8 for speech-to-speech tasks per GPU.

Table 9: **Multi-task Training Weights and Prompt Probabilities**

| Task | Weight | Prompt Probability |
|---|---|---|
| Speech Enhancement | 3.0 | 0.1 |
| Speaker Separation | 3.0 | 0.9 |
| Noise Isolation | 1.0 | 0.1 |
| Acoustic Matching | 1.0 | 0.9 |
| Speech Isolation | 1.0 | 0.1 |

Both stages use a shifted cosine noise schedule (scale=0.5) (Hoogeboom et al., 2023; Lovelace et al., 2024a) with sigmoid loss weighting (bias=-2.5) (Hoogeboom et al., 2024), mixed precision (bfloat16), and distributed data parallel (DDP) training.

### E.3 SAMPLING CONFIGURATION

We use the SDE-DPM-Solver++(2M) as described in Lu et al. (2022) for sampling. We utilize 256 inference steps with a schedule that is linear in logSNR. For speech-to-speech tasks, we utilize classifier-free guidance (Ho & Salimans, 2022) with a strength of 1.5. For zero-shot TTS we use guidance scale of 3.0 for the transcript and prompt conditioning information. For speech editing, we use a guidance scale of 2.0 for the transcript and prompt.

For zero-shot TTS and speech editing, our non-autoregressive approach requires determining the output duration before generation. We estimate this by first computing the speaking rate (phones per second) from the reference speech prompt. For zero-shot TTS, we then multiply this rate by the phoneme count of the target transcript to determine the output duration. For speech editing, we preserve the original duration for unedited regions and apply the same rate-based estimation for edited segments. We found this simple duration modeling approach sufficient for maintaining natural speaking rates aligned with the reference speaker's style.

For task composition, we can control the guidance strength in the same way. Higher guidance values enforce stronger conditioning at the cost of potentially conflicting with the other task. We use a scale of 1.5 in our composition experiments. We find that TTS guidance is only necessary for resolving details in modest-to-high SNR regimes, so we enable it for logSNR ranges greater than -1.0 (Kynkäänniemi et al., 2024).

## F SOURCE AUDIO CONDITIONING ABLATION

Table 10: **Source Audio Conditioning Ablation.** We train an ablation model that conditions on the source sequence vectors with a cross-attention mechanism instead of our frame-wise mixing.

| Model | PESQ ↑ | MCD ↓ | SpBS ↑ | WER ↓ |
|---|---|---|---|---|
| Noisy Source Audio | 1.12 | 11.22 | .888 | 3.3 |
| SpeechOp-Small (Cross-attention) | 1.18 | 15.4 | .751 | >100 |
| w/ chunking | 1.88 | 4.98 | .900 | 9.6 |
| SpeechOp-Small (Framewise-Mixing) (Ours) | 1.96 | 4.86 | .902 | 8.8 |

Using 12-layer models, we compare our framewise mixing strategy against a cross-attention based approach for conditioning on source audio. Table 10 shows that the cross-attention variant fails catastrophically when processing sequences other than its 5-second training length (WER > 100%). Even with explicit padding and chunking to account for this, it shows degraded performance across all metrics. In contrast, our framewise mixing approach generalizes naturally to arbitrary sequence lengths while achieving better quality (PESQ 1.96 vs 1.88), lower distortion (MCD 4.86 vs 4.98),

and improved content preservation (WER 8.8% vs 9.6%). These results suggest that framewise mixing provides a more robust foundation for speech-to-speech processing, likely due to the explicit frame-level correspondence between source and target audio.

## G  TASK COMPOSITION DERIVATION

Here we present the detailed derivation of our task composition approach. Our goal is to estimate the conditional score function $\nabla_{\mathbf{z}_t} \log p(\mathbf{z}_t|y, w)$, where $\mathbf{z}_t$ is the noisy latent, $y$ is the noisy source audio, and $w$ is the text transcript.

Starting with Bayes' rule, we can decompose the joint conditional probability:

$$
\begin{aligned}
\nabla_{\mathbf{z}_t} \log p(\mathbf{z}_t|y, w) &= \nabla_{\mathbf{z}_t} \log \frac{p(y, w|\mathbf{z}_t)p(\mathbf{z}_t)}{p(y, w)} \\
&= \nabla_{\mathbf{z}_t} \log p(y, w|\mathbf{z}_t) + \nabla_{\mathbf{z}_t} \log p(\mathbf{z}_t) - \nabla_{\mathbf{z}_t} \log p(y, w) \\
&= \nabla_{\mathbf{z}_t} \log p(y, w|\mathbf{z}_t) + \nabla_{\mathbf{z}_t} \log p(\mathbf{z}_t),
\end{aligned}
\tag{7}
$$

where we drop the term $\nabla_{\mathbf{z}_t} \log p(y, w)$ as it is independent of $\mathbf{z}_t$.

We introduce a conditional independence assumption: given the noisy latent $\mathbf{z}_t$, the textual transcript $w$ is independent of the noisy source audio $y$. That is:

$$
p(y, w|\mathbf{z}_t) = p(y|\mathbf{z}_t)p(w|\mathbf{z}_t)
\tag{8}
$$

This assumption is reasonable at modest-to-high signal-to-noise ratios where the latent representation effectively captures the salient information from both modalities. Substituting Equation equation 8 into Equation equation 7:

$$
\begin{aligned}
\nabla_{\mathbf{z}_t} \log p(\mathbf{z}_t|y, w) &= \nabla_{\mathbf{z}_t} \log p(y|\mathbf{z}_t)p(w|\mathbf{z}_t) + \nabla_{\mathbf{z}_t} \log p(\mathbf{z}_t) \\
&= \nabla_{\mathbf{z}_t} \log p(y|\mathbf{z}_t) + \nabla_{\mathbf{z}_t} \log p(w|\mathbf{z}_t) + \nabla_{\mathbf{z}_t} \log p(\mathbf{z}_t).
\end{aligned}
\tag{9}
$$

For the term $\nabla_{\mathbf{z}_t} \log p(w|\mathbf{z}_t)$, we can apply Bayes' rule again. Following the classifier-free guidance approach of Ho & Salimans (2022), this can be expressed in terms of conditional and unconditional TTS score functions:

$$
\nabla_{\mathbf{z}_t} \log p(w|\mathbf{z}_t) = \nabla_{\mathbf{z}_t} \log p(\mathbf{z}_t|w) - \nabla_{\mathbf{z}_t} \log p(\mathbf{z}_t).
\tag{10}
$$

Substituting Equation equation 10 into Equation equation 9, and noting that $\nabla_{\mathbf{z}_t} \log p(y|\mathbf{z}_t) + \nabla_{\mathbf{z}_t} \log p(\mathbf{z}_t) = \nabla_{\mathbf{z}_t} \log p(\mathbf{z}_t|y)$, we obtain:

$$
\begin{aligned}
\nabla_{\mathbf{z}_t} \log p(\mathbf{z}_t|y, w) &= \nabla_{\mathbf{z}_t} \log p(\mathbf{z}_t|y) + (\nabla_{\mathbf{z}_t} \log p(\mathbf{z}_t|w) - \nabla_{\mathbf{z}_t} \log p(\mathbf{z}_t)) \\
&\approx \mathbf{s}_\theta^{\text{enh}}(\mathbf{z}_t|y) + (\mathbf{s}_\theta^{\text{tts}}(\mathbf{z}_t|w) - \mathbf{s}_\theta^{\text{tts}}(\mathbf{z}_t)),
\end{aligned}
\tag{11}
$$

where $\mathbf{s}_\theta^{\text{enh}}(\mathbf{z}_t|y)$ and $\mathbf{s}_\theta^{\text{tts}}(\mathbf{z}_t|w)$ represent the score networks for enhancement and TTS tasks, respectively.

This derivation shows how our approach naturally combines the enhancement and TTS score functions while avoiding conflicts between their unconditional priors. The enhancement term guides the denoising process while the TTS term provides content alignment through classifier-free guidance.

## H  LLM USAGE

We used large language models for copyediting and revising the wording; all claims and arguments were drafted and verified by the authors.

