# OpenReview forum: "SpeechOp: Inference-Time Task Composition for Generative Speech Processing"
_ICLR.cc/2026/Conference — ICLR 2026 Poster_

### Official Review · Reviewer_2PAD · 2025-10-28

**Soundness:** 4
**Presentation:** 4
**Contribution:** 4
**Rating:** 8
**Confidence:** 4

**Summary:**

This paper introduces SpeechOp, a unified and versatile speech processor that leverages a pretrained DiT-based TTS model to perform a broad range of speech tasks (e.g., enhancement, separation, voice cloning) at inference time. The authors note that TTS models benefit from abundant in-the-wild data, S2S models typically rely on limited paired datasets, constraining their generative fidelity and data efficiency. To bridge this gap, SpeechOp employs a two-stage training paradigm, beginning with TTS pretraining followed by multi-task fine-tuning across multiple speech operations. Furthermore,  the paper proposes TC-CFG, a principled approach that fuses different modalities to improve the model performance at inference time. Experimental results demonstrate consistent improvements in both generative quality and task flexibility, validating SpeechOp as a strong step toward a universal, generative speech processor.

**Strengths:**

- The paper is clearly written and effectively motivates the need for unifying TTS and S2S paradigms.
- Empirical results convincingly demonstrate that initializing S2S models from a pretrained TTS backbone accelerates convergence and enhances performance.
- The proposed TC-CFG mechanism is conceptually well-founded and illustrated through a clear 1D toy example, highlighting its advantage over simple score averaging.
- The proposed framework effectively leverages different modalities to boost the task performance.
- The extensive experiments across TTS, enhancement, and separation tasks collectively validate the versatility and efficacy of SpeechOp.

**Weaknesses:**

(minor suggestion) Including the task embedding component in Figure 3 would further clarify how task conditioning is integrated across modalities.

**Questions:**

- How much additional training time does SpeechOp require compared to the standalone TTS baseline?
- Can the proposed TC-CFG mechanism be generalized to other modalities or tasks (e.g., direct application to TTS or voice conversion scenarios)?

---

> ### Author Response · Authors · 2025-11-26
>
> We thank the reviewer for their support and positive assessment of our contributions.
>
> **On Including Task Embedding in Figure 3**
>
> Thank you for this suggestion. We will update Figure 3 to explicitly show how the task embedding conditions both the Audio Encoder and Diffusion Transformer.
>
> **On Additional Training Time**
>
> The multi-task fine-tuning stage adds 200K iterations beyond the 400K TTS pretraining iterations. However, this fine-tuning procedure is significantly more efficient than training specialized speech processing models from scratch. TTS initialization provides 4-8x faster convergence for S2S tasks compared to random initialization (Figure 2). If one views TTS training as a sunk cost, since TTS models are already being developed for various applications, then the multi-task fine-tuning stage is an efficient way to extend a TTS model into a unified speech processor with diverse S2S capabilities, while also improving the underlying TTS performance.
>
> **On TC-CFG Generalization**
>
> Yes, TC-CFG can be applied to other modalities and tasks. The formulation we present (Eq. 4) is general and is not restricted to the tasks we explore or even the speech domain. It incorporates the discriminative signal from any conditional diffusion model to guide generation of some base diffusion model. For voice conversion, one could use the discriminative signal from a speaker-conditioned model to guide a base model that preserves content. In preliminary experiments, we explored other combinations such as composing voice cloning with enhancement (cloning a target speaker's voice while improving the acoustics) and found our approach to be effective. We focused on the settings in the paper due to the availability of clear benchmarks and evaluation protocols, but we believe TC-CFG has broad applicability. We will add discussion of possible extensions and additional applications to the final version.

---

> > ### Comment · Reviewer_2PAD · 2025-11-27
> >
> > Thank you for the detailed explanations. I will maintain my original score.

---

### Official Review · Reviewer_omQ3 · 2025-10-28

**Soundness:** 2
**Presentation:** 3
**Contribution:** 3
**Rating:** 4
**Confidence:** 4

**Summary:**

The paper proposes a multi-task training method that jointly learn on TTS and speech processing tasks. The paper also proposes a test time composition method using CFG to better guide the speech processing task.

**Strengths:**

1. While multi-tasking learning is not new, the paper propose some novel design choices that let TTS and speech processing tasks to be trained more seamlessly in a diffusion model.
2. The inference time task composition method uses text conditioning CFG to additionally guide the speech processing task and seems to have good results.
3. The paper is in general written clearly and easy to follow.

**Weaknesses:**

1. The competitors compared in the evaluation section are in general too few and not SoTA. For example, in TTS all models compared are from 2024. There are many new TTS models (and some good open sourced ones like Higgs Audio) in 2025. I think the evaluation section need to be updated with results comparing against more recent methods. The same for speech editing where only one work is compared against.
2. While the paper performed a simple ablation on the task composition, it didn't run any ablation on the design choices of the training methods and model architecture.

**Questions:**

None

---

> ### Author Response · Authors · 2025-11-26
>
> We thank the reviewer for their constructive feedback.
>
> **On Baseline Comparisons and SOTA**
>
> We appreciate this concern and wish to clarify the goals of our evaluation. The goal of our TTS evaluation is to validate that our model is competitive with recent systems of comparable scale, not to claim state-of-the-art TTS. Our primary contributions are: (1) demonstrating 4-8x faster convergence through TTS to S2S transfer (Figure 2), (2) our novel TC-CFG formulation that significantly outperforms score averaging (38% relative WER reduction, Table 6), and (3) the ITC pipeline achieving state-of-the-art content preservation.
>
> The model the reviewer references (Higgs Audio) has 5.8B parameters and is trained on over 10M hours of speech data. This makes it roughly 14x larger with over 200x more training data than SpeechOp (419M parameters, ~45k hours). Comparing against such models would primarily reflect differences in scale rather than our methodological contributions.
>
> Within our parameter and data regime, SpeechOp exhibits strong performance compared to recently published systems. We match or exceeds CLaM-TTS (584M, ~56k hrs) and XTTS (482M, ~17k hrs) on most metrics. We also surpass VoiceCraft (830M, ~69k hrs) in intelligibility and subjective quality. Our evaluation also demonstrates for the first time that multi-task training has the potential to improve TTS performance (Table 1) in terms of speaker identity metrics.
>
> For speech editing, VoiceCraft (830M parameters, \~69k hrs) is the primary recent baseline with publicly available code and model weights. Despite being significantly larger (\~2x) and trained on more data than SpeechOp (419M, ~45k hrs), we outperform it across all subjective metrics (Table 2). We would be glad to hear if the reviewer has specific suggestions for additional baselines we may have overlooked.
>
> **On Ablations**
>
> We provide ablations on our core novel contributions:
>
> * TTS initialization vs. random initialization (Figure 2): 4-8x faster convergence for speech processing tasks
> * Task composition method (Table 6): TC-CFG vs. TC-Avg shows significant improvements in content preservation and signal fidelity
> * Source audio conditioning (Appendix F): Framewise mixing vs. cross-attention demonstrates that our audio conditioning approach is critical for strong performance on S2S tasks.
>
> We focused our ablations on these contributions rather than standard architectural choices. Importantly, adopting a standard DiT architecture is a deliberate strength of our approach: we use the same DiT backbone employed in recent large-scale TTS systems like DiTTO-TTS. This means our multi-task fine-tuning strategy and TC-CFG composition method can be directly applied to stronger models as they emerge. Our findings are not tied to a specific architecture but generalize to the broader class of DiT-based TTS models, making our contributions immediately applicable as the field advances and models scale.

---

### Official Review · Reviewer_SsZ8 · 2025-10-29

**Soundness:** 3
**Presentation:** 3
**Contribution:** 3
**Rating:** 8
**Confidence:** 3

**Summary:**

This goal of this paper is to develop a speech model that can perform some speech to text and speech to speech tasks by starting from a diffusion based model that is trained only for TTS initially. The argument is the availability of relatively larger scale data for TTS task while lack of sufficient paired data for speech denoising, speaker separation, speech enhancement, etc. tasks. In addition to multitask training, the model allows task composition during inference time, such as Text-to-Speech + Enhancement, Text-to-Speech + Separation in one model. This is achieved by a method called the implicit task composition (ITC) where a Whisper based ASR transcripts are used in place of  text component to guide the proposed SpeechOp model's speech enhancement process. ITC does not only lead to a model capable of handling different tasks but also improves the TTS task performance that the model was initially trained on. The TTS model is based on diffusion modeling on DAC based latent speech features. In addition to text and audio features, the model hasa a learnable Task Embedding that conditions model behavior. Once the basic TTS model is trained the multitask training consists of two stages. In the first stage, TTS and S2S tasks are jointly trained. In the diffusion training step, classifier free guidance method is used. For the multitask method the training objective is a weighted sum of the speech enhancement and the TTS losses. The system is trained several open source datasets and all speech signals are upsample to 48kHz. Experiments compared the performance on both individual tasks and task composition performance. Experiments claim competitive audio quality, significantly outperforming SepFormer based speech separation  across all datasets. Especially, experiments suggest that the text guidance is an important part of the proposed SpeechOp.

**Strengths:**

Originality
+ The paper proposes a text guided multitask modeling for several speech tasks in one model that was initially trained for TTS. Thus the model can leverage the relatively high-resource TTS task in other speech related tasks such as speech separation, and inclusion of the text guidance allow for inference time task composition that can expand the model capabilities.

Quality
+ Experimental results suggest that the generated audio quality is competitive as compared to audio reconstruction with a HiFiGAN2 vocoder. In speech separation, the paper outperforms the Sepformer based baselines.

Clarity
+ Mostly clearly written.
+ Datasets used in the experiments are open source. Some experimental settings are provided in the appendix.

Significance
+ Since the SpeechOp paper is a multitask speech model capable of handling several speech tasks, it is relevant to various subsets of speech researchers.

**Weaknesses:**

The following points are mostly about the clarity of the content or the quality of the experimentation.

- One point is that is not very clear is how critical is the task embedding in the system if Fig. 3. or whether given the complex interactions between diffusion latents and text encodings through cross attention, it is not clear whether the model can leverage the task embedding efficiently.

- In some experiments, SpeechOp results are presented in (no transcript) and (ground truth transcript) versions. For completeness, it could have been useful to see the SpeechOp results with the ASR transcriptions (e.g. in Table 4).

- In Table 3, there is a reference to Speaker Personalization but I could not see the explanation of how this is achieved. Could you please explain?

**Questions:**

1. Please discuss the effectiveness of the task embeddings in the SpeechOp model more clearly.

2. In some experiments, SpeechOp results are presented in (no transcript) and (ground truth transcript) versions. For completeness, it could have been useful to see the SpeechOp results with the ASR transcriptions (e.g. in Table 4). Could the authors please comment on that?

3. In Table 3, there is a reference to Speaker Personalization but I could not see the explanation of how this is achieved. Could you please explain?

4. (Minor typo) Section 5. Speech-to-Speech Pathway: us a complex alignment mechanism -> *use* a complex alignment mechanism

---

> ### Author Response · Authors · 2025-11-26
>
> Thank you for your thoughtful review and for your support. We address your questions below. Please let us know if we can provide any additional information during the discussion.
>
> **On the Task Embedding**
>
> The original Diffusion Transformer (DiT) paper explored class-conditional image generation on ImageNet and introduced a class embedding to guide generation towards some target class (e.g. Tiger vs Lion). In our work, we adopt the same approach to condition both the Audio Encoder and Diffusion Transformer on the desired speech processing task with a learnable task embedding. We chose to adopt the same approach as the original DiT paper for simplicity. Our successful multi-task performance across diverse tasks demonstrates the task embedding lets the network learn task-specific behavior. We include qualitative samples in the supplementary material showing the different generations achieved for the same audio input with different task specifications (e.g. separation versus enhancement). This shows the effect of the task embedding on the final generated output.
>
> However, we agree that more expressive approaches for task conditioning may prove more effective. We will add discussion clarifying our decision to adapt the class embedding approach from the original DiT paper in the revision and mention further exploration of task conditioning/specialization as a potential direction for future work.
>
> **On ASR Transcription Results**
>
> Thank you for this suggestion. Our ITC pipeline uses WhisperX transcriptions from noisy audio, achieving a WER of 2.9% for enhancement (Table 3) compared to the 3.3% WER obtained from the noisy source audio.
>
> As you noted, for speaker separation, we conduct our task composition evaluation with the gold transcripts to present an upper bound for settings with reliable transcription. For that task, we evaluate on standard academic benchmarks (e.g., WSJ0-2Mix, LibriMix) which consist of fully overlapping synthetic speech mixtures (examples in the supplementary) to match the evaluation protocol of prior work (e.g. our Sepformer baseline). In this setting, standard ASR models struggle since speech from both speakers overlaps throughout the entire utterance.
>
> However, in more realistic conversational scenarios, speakers typically take turns with only ocassional overlap. In such cases, off-the-shelf ASR with speaker diarization can reliably transcribe the non-overlapping regions, providing timestamps and transcripts that can guide SpeechOp's separation during overlapping segments. Our gold transcript results (Table 4, Table 5) demonstrate that textual guidance significantly improves separation quality (e.g., WER drops from 11.1% to 5.5%) and subjective quality (MOS from 3.57 to 3.92), suggesting ITC could be very beneficial for separation in realistic settings.
>
> We will clarify this limitation of our speaker separation evaluation setting and discuss extensions to realistic conversational scenarios in the revision.
>
>
> **On Speaker Personalization**
>
> We apologize for the lack of clarity regarding this composition. For Speaker Personalized Enhancement, we compose enhancement with voice cloning by providing a clean voice sample from the target speaker alongside the noisy audio. Voice cloning is performed using the inpainting capabilities of our base TTS model (similar to VoiceBox). By providing a clean speech reference, the model can better preserve speaker-specific characteristics during enhancement. We find that this improves signal fidelity (MCD, PESQ) and modestly reduces WER (2.4% vs 2.9%). We will clarify this task composition implementation in the revision.
>
> **Minor Typo**
>
> Thank you for your careful reading of our work. We will fix this in the final version.

---

> > ### Comment · Reviewer_SsZ8 · 2025-11-27
> >
> > Thank you for the detailed responses. I will maintain my original score.

---

### Official Review · Reviewer_vHfv · 2025-11-02

**Soundness:** 3
**Presentation:** 3
**Contribution:** 2
**Rating:** 4
**Confidence:** 4

**Summary:**

This paper presents SpeechOp, a multi-task latent diffusion model designed to transform pre-trained Text-to-Speech (TTS) models into a universal speech processor capable of performing and composing multiple speech-to-speech (S2S) tasks (e.g., enhancement, separation, voice cloning) at inference time. The key innovation lies in the Task Composition Classifier-Free Guidance (TC-CFG) mechanism and Implicit Task Composition (ITC) framework, which allow flexible inference-time combination of speech operations using text or ASR-derived guidance (e.g., from Whisper). The authors show that TTS pre-training accelerates convergence and improves downstream S2S performance, and that SpeechOp achieves state-of-the-art content preservation across tasks such as enhancement and separation.

**Strengths:**

1. The idea of repurposing pre-trained TTS models for versatile S2S operations via inference-time task composition is well-motivated, addressing the data scarcity challenge in generative S2S tasks.
2. The paper reports substantial improvements in speech enhancement and perceptual quality gains in speaker separation.

**Weaknesses:**

1. The idea of leveraging pre-trained diffusion-based TTS models as clean speech generators for downstream tasks such as denoising, enhancement, and separation has already been proposed in Voicebox, Audiobox, and Speechflow. The paper’s novelty mainly lies in its composition mechanism, rather than the overall concept of using TTS priors for multi-task speech processing.
2. While the composition strategy is interesting, the underlying model (latent diffusion with DiT backbone) largely follows existing designs (Voicebox, DiTTo-TTS, etc.). The architectural contribution is modest.
3. ITC’s success depends heavily on accurate ASR transcripts. The paper lacks robustness analysis under ASR noise or low-resource conditions. Maybe using some softer semantic supervision like SSL features instead of the explicit transcripts would mitigrate this issue.

**Questions:**

How does SpeechOp perform on cross-lingual or non-English speech tasks, given the ByT5 encoder and English-only datasets?

---

> ### Author Response · Authors · 2025-11-26
>
> We thank the reviewer for engaging with our work.
>
> **On Novelty and Prior Work**
>
> We appreciate the opportunity to clarify the distinction between our work and the prior works mentioned.
>
> Voicebox and Audiobox are primarily text-to-speech systems. While Voicebox does support transient noise removal (e.g. removing a doorbell or dog bark) this is done through masked infilling. This requires users to identify and mask corrupted segments in otherwise clean audio for regeneration. This is fundamentally different from the general speech processing tasks we address (enhancement, speaker separation, speaker isolation), which process arbitrarily degraded audio without requiring manual segmentation. Audiobox extends Voicebox to text-to-audio/sound generation but similarly does not address these speech processing tasks.
>
> SpeechFlow, as discussed in Section 3, proposes a dedicated pretraining scheme on speech data specifically designed for downstream speech generation tasks. In contrast, we demonstrate efficient transfer from existing TTS models. Given that TTS models are already being developed and deployed for various applications, our approach enables leveraging these models for speech processing without any additional pretraining costs.
>
> Critically, none of these works explore inference-time task composition. Our TC-CFG formulation enables composing independently trained task capabilities at inference time (e.g., combining enhancement with TTS guidance). This is a fundamentally different contribution from prior work.
>
> To summarize, our primary contributions are: (1) demonstrating 4-8x faster convergence through TTS-to-S2S transfer (Figure 2), (2) our novel TC-CFG formulation that significantly outperforms score averaging (38% relative WER reduction, Table 6), and (3) the ITC pipeline achieving state-of-the-art content preservation. These are unique to our work and were not explored in the prior works cited by the reviewer.
>
> **On Architecture**
>
> We agree our architecture follows standard practices (DiT backbone). However, this is an intentional design choice. By adopting established architectures, we can isolate and demonstrate the effectiveness of TTS to S2s transfer, our multi-task training procedure, and TC-CFG composition strategies. Our approach is directly applicable to other DiT-based TTS models such as Voicebox and DiTTO-TTS. We consider this generality to be a strength of our work.
>
> **On ASR Robustness**
>
> The reviewer raises a valid concern about ASR noise. However, our results demonstrate reasonable robustness to ASR errors. We observe that even when using Whisper transcripts from noisy source audio, ITC improves content intelligibility even over the original audio (WER 2.9% vs. 3.3%). This suggests TC-CFG effectively balances acoustic information with imperfect textual guidance. Additionally, TC-CFG provides a tunable guidance strength that allows practitioners to adjust the transcript influence based on the ASR quality. Transcript-conditioned models, on the other hand, lack this kind of flexibility.
>
> Incorporating SSL features is also an interesting direction and is directly compatible with our broader framework. TTS composition can similarly be applied to an SSL-conditioned enhancement model. We will add discussion of this potential direction to our final draft. One benefit of tunable TTS composition is that it enables adapting specialized ASR models for specific domains and applications at inference-time in a plug-and-play manner. An SSL-conditioned enhancement model would be limited to speech conditions included in SSL pretraining. Adapting it to new speech environments would require developing a new SSL backbone and re-training the diffusion model.
>
> **On Cross-Lingual Performance**
>
> Our current experiments use English-only training data. However, ByT5 is a multilingual language model that already supports 100+ languages. As a result, extending SpeechOp to other languages only requires the inclusion of multilingual training data. We will acknowledge such an extension as an important direction for future work.

---

### Meta-Review · Program_Chairs · 2026-01-04

**Summary:**

The paper presents a model for improving speech-to-speech (S2S) tasks, like enhancement, by improving a pre-trained TTS model to allow for multiple tasks and adding the ability to do task composition during inference. The reviewers commented positively on the novelty, the tasks tackled by the model, and the overall quality. And importantly, how it addresses some of the data issues related to S2S tasks.

Main issues that were raised:
1. Additional analysis on the utility of task embeddings.
2. Some missing experiments and results.
3. Outdated baselines for TTS.
4. Missing ablations on model architecture.

**Reviewer Concerns:**

1. The authors clarify that they adopted the architecture based on DiT implementation, and added qualitative examples with various task embedding. Improving task embedding will be handled in future work.
2. For multitalker, the authors argue that typical ASR models struggle, and that full overlap happens only occasionally. Nonetheless, results with ASR transcripts would have made the paper more complete.
3. The authors argue that the goal is not to claim SOTA TTS, which is fair.
4. Included additional ablations.

**Reviewer Scores:**

Reviewer vHfv: 4 -> 4 [marked as AI review]

Reviewer SsZ8: 8 -> 8

Reviewer omQ3: 4 -> 6

Reviewer 2PAD: 8 -> 8

---

### Decision · Program_Chairs · 2026-01-26

Accept (Poster)